# Cerebral Glutamate Regulation and Receptor Changes in Perioperative Neuroinflammation and Cognitive Dysfunction

**DOI:** 10.3390/biom12040597

**Published:** 2022-04-18

**Authors:** Yan Zhang, John-Man-Tak Chu, Gordon-Tin-Chun Wong

**Affiliations:** Department of Anaesthesiology, LKS Faculty of Medicine, The University of Hong Kong, Pokfulam, Hong Kong SAR, China; u3005066@connect.hku.hk (Y.Z.); jmtchu@hku.hk (J.-M.-T.C.)

**Keywords:** neuroinflammation, AMPA receptor, NMDA receptor, glutamate, microglia, astrocyte, perioperative, anesthetic, neurotransmission, excitotoxicity

## Abstract

Glutamate is the major excitatory neurotransmitter in the central nervous system and is intricately linked to learning and memory. Its activity depends on the expression of AMPA and NMDA receptors and excitatory amino transporters on neurons and glial cells. Glutamate transporters prevent the excess accumulation of glutamate in synapses, which can lead to aberrant synaptic signaling, excitotoxicity, or cell death. Neuroinflammation can occur acutely after surgical trauma and contributes to the development of perioperative neurocognitive disorders, which are characterized by impairment in multiple cognitive domains. In this review, we aim to examine how glutamate handling and glutamatergic function are affected by neuroinflammation and their contribution to cognitive impairment. We will first summarize the current data regarding glutamate in neurotransmission, its receptors, and their regulation and trafficking. We will then examine the impact of inflammation on glutamate handling and neurotransmission, focusing on changes in glial cells and the effect of cytokines. Finally, we will discuss these changes in the context of perioperative neuroinflammation and the implications they have for perioperative neurocognitive disorders.

## 1. Introduction

It is becoming apparent that systemic circulating factors can pass through a dysfunctional blood–brain barrier (BBB) and have a profound impact upon brain homeostasis, aging, and neurodegeneration [1]. The systemic immune response to endogenous or exogenous triggers releases substances that can, in turn, initiate immune and inflammatory reactions in the central nervous system (CNS). Neuroinflammation, oxidative stress, and excitotoxicity are associated with several neurological disorders, such as Alzheimer’s disease (AD) [2,3,4], Parkinson’s disease [5,6,7], and multiple sclerosis [8,9,10]. Neuroinflammation can cause disruptions to synaptic transmission and glial and neuronal dysfunction that contribute to cognitive impairment. Among these changes are alterations to the release, uptake, and clearance of glutamate, as well as changes in the functions and subunit composition of its receptors [11]. Glutamate, being the major excitatory neurotransmitter in the CNS, is primarily released from presynaptic vesicles and acts on different types of postsynaptic glutamate receptors [12,13].

Unlike the development of neurodegenerative diseases that occurs without a single identifiable trigger or event, perioperative neurocognitive disorders (PNDs) are clearly linked to surgery. This group of neurological disorders encompasses a range of conditions from acute delirium to more sustained postoperative cognitive dysfunction. Tissue trauma from surgery causes the release of substances that can overwhelm the immune system and set up an inflammatory response in the brain. Patients are usually exposed to anesthetic agents that also have intrinsic effects on neurotransmitters and neurotransmission. Accumulating evidence indicates that the overactivation of the inflammatory response and the change in glutamate metabolism contributes to the development of PNDs [14,15].

The purpose of this review is to outline how glutamatergic neurotransmission is affected by perioperative neuroinflammation. We will first summarize the current data regarding glutamate receptor configuration and trafficking, and glutamate release and handling in the CNS. We will then provide a summary of the effects of peripheral immune cells and cytokines on glial cells and how they alter normal glutamate regulation. Finally, we will outline the experimental evidence on the effects of commonly used anesthetic agents and surgical trauma on glutamate in the CNS and some potential treatments.

## 2. Glutamate Receptors

Glutamate receptors can be broadly divided into ionotropic and metabotropic types. Ionotropic receptors are coupled with ion channels to form receptor–channel complexes that mediate fast signal transmission; these include the α-amino -3 hydroxy-5 methyl-4 isoxazole propionic acid receptors (AMPARs), *N*-methyl-d-aspartate receptors (NMDARs), and the kainate receptors (KARs). Metabotropic glutamate receptors are coupled with G-proteins on cell membranes, and include mGluR1 to mGluR8 types [16,17]. GluA1 to Glu4 AMPARs are tetrameric assemblies of two dimers encoded by the four GRIA (GRIA1 to GRIA4) genes. The four main tetrameric complexes are composed of GluA1/2 and GluA2/3 heteromers and GluA1 homopolymers. GluA1/A2 heteromers are the dominant AMPARs in the CA1 hippocampus, with around 80% of synaptic and more than 95% of somatic extra-synaptic receptors of this type, with the remainder being the GluA2/GluA3 heteromers. GluA4 mainly appears during embryonic development [18,19]. Calcium-permeable AMPARs (mainly consisting of a GluA2-lacking AMPAR) may emerge under some pathological conditions, such as status epilepticus, glaucoma, and neuroinflammation. This GluA2-lacking AMPAR has a linear current/voltage curve and is permeable to Ca^2+^ [20,21,22,23].

The NMDAR is a heterotetramer composed of NR1 to NR3 subunits, with NR2 having four subtypes (NR2A, NR2B, NR2C, and NR2D) and NR3 having two subtypes. Under normal conditions, NMDARs are blocked by magnesium ions; they are activated after postsynaptic depolarization and the removal of these ions [24]. However, the overactivation of NMDARs can lead to neuronal excitotoxicity, cell apoptosis, and cell death through the activation of calcium ion-mediated intracellular pathways [25].

Metabotropic glutamate receptors exist both in the CNS and peripherally and are mainly expressed in neurons and glial cells in proximity to the synaptic cleft [26,27]. Metabotropic glutamate receptors (mGlus) are a family of G-protein-coupled receptors activated by the glutamate neurotransmitter. The family has eight molecular clones termed metabotropic glutamate receptor 1–8 (mGlu1–8) [28]. mGluRs can be divided into three groups (Group I–III mGlus). Group I includes mGlu1 and mGlu5, Group II includes mGlu2–3, and Group III includes mGlu4 and mGlu6–8 [29]. mGlus play crucial roles in the modulation of neuronal excitability, synaptic plasticity, and the release of neurotransmitters [30]. Most metabotropic glutamate receptors are located presynaptically, except for the Group I (mGluR1 and mGluR5) receptors [31]. These Group I receptors can increase the activity of NMDARs and induce excitotoxicity [32]. In addition, mGluR5 can mediate experience-dependent NMDA subunit switching [33].

## 3. Glutamate Release and Handling

Glutamate can be released by vesicular or non-vesicular release mechanisms. Under physiological conditions, synaptic release is primarily via the vesicular mode. When an action potential reaches the terminal, an influx of Ca^2+^ triggers the exocytosis of glutamate vesicles. Ca^2+^ can then bind to synaptotagmin, causing it to bind to a complex composed of SNARE and Sec1/Munc18-like (SM) proteins that mediate membrane fusion during exocytosis, thus promoting the release of neurotransmitters [34]. The non-vesicular mechanism occurs under pathological conditions and involves anion channel release, the reverse efflux of glutamate, and xC-system release [35]. The anion channels are mainly located on astrocytes and can release glutamate if the astrocytes become swollen, as seen in ischemia-reperfusion injury [36]. The efflux of glutamate is mediated through excitatory amino transporters (EAATs) located on endothelial cells and glial cells. EAATs help with the uptake of glutamate; the BBB can function in efflux mode to selectively move glutamate from the brain to the blood. The glial cells lining the BBB take up the glutamate and release it into the proximity of endothelial cells, promoting efflux. [37]. Glutamate transport in the outward direction is termed reverse transport. The reverse transport of glutamate occurs both in neurons and astrocytes. Glutamate can be released extracellularly by reverse transport when the neurons’ extracellular Na^+^/intracellular K^+^ levels decrease or when the intracellular Na^+^/extracellular K^+^ levels increase. This process is mediated by the electrochemical gradient of co- and counter-transported ions produced by the glutamate transporter EAAC1 (EAAT3) [38,39]. In astrocytes, reverse transport happens in some extreme situations; for example, after ATP depletion, the membrane gradients collapse, glutamate uptake ceases, and the efflux of glutamate occurs via reverse transport [40]. In addition, astrocytes can release glutamate through other modalities such as exocytosis, hemichannels, anion transporters, and P2X receptors [41]. The system xC- is mainly located in glial cells and is responsible for exchanging glutamate with cystine, a substrate used for the synthesis of the antioxidant molecule glutathione (GSH). In many brain regions, the xC-system acts as the main source of intracellular cystine by exporting glutamate extracellularly. Extracellular glutamate can inhibit the xC-system and contribute to the depletion of GSH, which can lead to oxidative glutamate toxicity. Inflammatory cytokines, including TNF-a and IL-1β, can upregulate the xC-system. Increased xC expression can have neuroprotective effects or excitotoxic side effects in different animal models [42].

Glutamate would accumulate in the brain if not for the transporter proteins that remove it from the extracellular fluid to maintain low extracellular concentrations. Extracellular glutamate is controlled by a family of plasma membrane enzymes called EAATs. EAAT 1–5 are encoded by the SLC1A3, SLC1A2, SLC1A1, SLC1A6, and SLC1A7 genes, respectively [43]. EAAT1/2 are considered glial transporters and are widely found in the cerebellum and forebrain, while EAAT3/4 are considered as neuronal transporters and are widely distributed in the forebrain, spinal cord, and cerebellum [44,45,46]. Glutamate transporters are important for the termination of excitatory signals, glutamate recycling, and the prevention of excitotoxic injury [47]. GLAT1 (EAAT2) is expressed at high levels in brain astrocytes and at lower levels in neurons. Three variants of GLT1 exist (GLT1a, GLT1b, and GLT1c); GLT-1a is the only glutamate transporter subtype identified in axon terminals and contributes significantly to glutamate uptake into excitatory terminals [48,49].

## 4. Regulation of Glutamatergic Neurotransmission by Glial Cells

### 4.1. Microglia

Microglia are innate immune cells in the brain parenchyma that have similar actions to circulating macrophages [50]. In addition to their role in mediating immune responses in the CNS, microglia also provide nutrition to neurons and can respond dynamically to changes in neuronal activity [51]. Microglia can prune developing synapses and regulate synaptic plasticity and function. The dysfunction of microglia–synapse interactions can lead to synapse loss and neurodegenerative disease [52]. While in their surveillance state, microglia constantly scan the local microenvironment, but once activated, they can exhibit different morphologies and their functions can vary from being pro-inflammatory to anti-inflammatory [53]. They can become activated in the perioperative period through a series of peripherally initiated processes. Traumatized tissues in the body can release damage-related molecular patterns (DAMPs), such as high molecular group box I protein (HMGB1). When these DAMPs are combined with Toll-like receptors (TLRs) and receptors for advanced glycosylation end products (RAGEs) on the surface of bone marrow-derived monocytes (BMDMs), the nuclear translocation of NF-κB occurs, resulting in the increased expression of cytokines [54,55,56]. The secreted pro-inflammatory cytokines can act on BMDMs through a positive feedback loop to further promote the translocation of NF-κB and the release of cytokines [57]. Pro-inflammatory cytokines can upregulate the cyclooxygenase 2 isozyme (COX-2) [58]. 

Under normal conditions, the BBB can prevent the entry of harmful substances into the brain. However, the presence of peripheral pro-inflammatory cytokines damages the BBB via the action of COX2 and matrix metalloproteins (MMPs) in the endothelial cells. This allows pro-inflammatory cytokines together with BMDMs to migrate into the CNS, which in turn leads to the activation of microglia [59]. Inflammatory cytokines can promote the release of glutamate [60,61], and the glutamate that is released from activated microglia in turn stimulates glutamate receptors on the microglia to further release cytokines [62]. Thus, activated microglia can display a positive feedback loop to amplify the further release of cytokines and discharge a large amount of glutamate into the extra-synaptic space [63] (Figure 1). The degradation of extracellular ATP alleviates glutamate-induced inhibition of microglial proliferation [64]. Metabotropic glutamate receptors (mGluR2/5) are expressed on microglia, and when the mGluR2 receptor is activated, they can enhance the release of inflammatory cytokines, including TNFα, glutamate, and nitric oxide (NO), leading to neurotoxicity [65,66]. On the other hand, the activation of mGluR5 seems to have an opposite effect on neuroprotection [67].

### 4.2. Astrocytes

Astrocytes can provide nutritional support, maintain synaptic homeostasis, regulate synaptic pruning, and participate in neural signal transduction. They play essential roles in oxidative stress and the regulation of glutamate metabolism and cycling [68,69,70]. Microglia can initiate an immune response in the CNS and subsequently activate astrocytes [71]. The increased expression of glial fibrillary acidic protein (GFAP) is a marker of astrocyte activation. Glia cells share some common transcriptional pathways after neuroinflammation occurs, such as the NF-κB pathway. The nuclear translocation of NF-κB is boosted by TNF-α, interleukin (IL)-1b, and IL-17. In addition, sphingolipids, such as sphingosine 1-phosphate (S1P) and lactosylceramide (LacCer), can also trigger NF-κB translocation [72]. The sodium-dependent glutamate transporters EAAT2 are present on astrocytes [73]. Dysfunctional glutamate transporters and increased extracellular glutamate levels can cause neuronal injury. EAAT2 can reduce excess glutamate levels in the synaptic cleft to reduce excitotoxicity [74]. More than 90% of glutamate is cleared by the type 2 EAAT (Figure 1). EAAT2 (termed glutamate transporter I (GLT-1) in rodents) is the major amino transporter of glutamate in the CNS. Glutamate can be converted into glutamine, which is then released and taken up by neurons and transported to synaptic vesicles through glutamate transporters (VGLUT1-3) to complete the cycle of glutamine metabolism [73,75]. Astrocytes express many immune-derived receptors, including those for cytokines, chemokines, and complement proteins, and activation by these factors can recruit macrophages to enter the CNS [76,77]. β-catenin, a transcriptional co-activator in the Wnt/β-catenin pathways expressed in astrocytes, can positively regulate EAAT2 at the transcriptional level in progenitor-derived astrocytes by partnering with T cell factor 1 and 3 [78]. TNFα can increase the release of glutamate and decrease EAAT2 protein expression in astrocytes [79,80].

### 4.3. Oligodendrocytes

Oligodendrocytes and astrocytes can mutually affect each other during neuroinflammation. Oligodendrocytes secrete pro-inflammatory cytokines to induce NF-κB signaling and pro-inflammatory functions in astrocytes [72,81]. The receptors on oligodendrocytes respond to the inflammatory stimuli secreted by the astrocytes. Activated astrocytes promote the myelination and apoptosis of oligodendrocytes via TNFα, Fas ligand (FasL), and glutamate secretion [82,83,84]. Oligodendrocytes have effects on excitatory neurotransmission. Oligodendrocytes are highly vulnerable to AMPA and kainate receptor-mediated toxicity. AMPA and kainate receptor-mediated excitotoxicity contributes to demyelination and axonal injury in mature oligodendrocytes. Glutamate regulation has a potential neuroprotective strategy, as evidenced by the deletion of GluA4 from mature oligodendrocytes in experimental autoimmune encephalomyelitis (EAE) [85,86]. The overactivation of NMDARs can impair myelin synthesis. Activated microglia release glutamate through the system xc- cystine-glutamate antiporter and block glutamate transporters in oligodendrocytes [87]. In addition, pro-inflammatory cytokines can also impair the clearance of glutamate by the EAATs on oligodendrocytes [88].

## 5. Regulation of Glutamate Neurotransmission by Peripheral Immune Cells

### 5.1. Macrophages

Macrophages can be recruited into the CNS by CCL2/CCR2 signals [89]. Many of these macrophages reside in the perivascular area and express glutamate transporters and both metabotropic and ionotropic receptors. Macrophages can secrete glutamate, thus increasing the excitotoxicity of the inflammatory environment [90]. In patients with neurodegenerative diseases such as AD, β-Amyloid protein can enhance the macrophage’s ability to produce more oxygen free radicals and glutamate [91]. In addition, it can induce NMDAR-mediated neurotoxicity by secreting glutamate [90]. The activated macrophage can contribute to spine loss in multiple sclerosis (MS) and EAE by secreting glutamate, inflammatory cytokines, free radicals, and MMPs [92].

### 5.2. T cells

T cells can balance glutamatergic and GABAergic neurotransmission in the CNS to decrease excitotoxicity and can attenuate astrocyte activity [93]. They express AMPA GluR3 subunits and NMDARs and respond to glutamate in a dose-dependent manner [94]. Low concentrations of glutamate can promote T cell adhesion and migration, whereas higher concentrations can act on AMPARs and NMDARs to stimulate proliferation and metabotropic glutamate (mGluRs) receptors to reduce cell apoptosis [95].

## 6. The Effects of Pro-Inflammatory Cytokines on Glutamatergic Function

### 6.1. Interleukin IL-1β 

IL-1β is a pro-inflammatory cytokine that can act on NMDARs to increase NMDA receptor-induced intracellular calcium increase, an action that can be abolished by IL-1 antagonists [96]. This cytokine can also inhibit NMDAR-mediated synaptic transmission by depressing the isolated NMDA-EPSP amplitude in the dentate gyrus [97]. IL-1β can also inhibit the uptake of glutamate by astrocytes [98]. 

### 6.2. IFN-γ

IFN-γ is produced by T lymphocytes [93] and can change the phenotype of astrocytes [11] to stimulate glutamate clearance [98]. It can also stimulate macrophages to secrete glutamate compounds, such as QUIN, and alter glutamatergic neurotransmission [99]. IFN-γ has been shown to enhance glutamate neurotoxicity through AMPARs; the IFN-γ receptor forms a CP-AMPA receptor complex and phosphorylates GluRl at serine 845 via the JAKT2/STAT1 pathway [100].

### 6.3. Interleukin 6 (IL-6)

IL-6 is a crucial component of the inflammatory response and has important roles in the immune and hematopoietic systems [101]. Experimentally, it has been shown to protect cultured hippocampal neurons from glutamate-induced cell death [102]. However, chronic IL-6 exposure disrupts neuronal function and may contribute to the pathophysiology associated with many neurological diseases [103]. IL-6 inhibits glutamate neurotransmitter release in the cerebral cortex accompanied by the stimulation of STAT3 tyrosine phosphorylation [104].

### 6.4. TNFα

Pro-inflammatory cytokines can regulate synaptic strength, and AMPARs play an important role in synaptic plasticity in this regard. TNFα can change the AMPAR subunits to have a significant effect on neurotransmission. [105]. Under physiological conditions, GluR2-containing AMPARs are not permeable to Ca^2+^ and are resistant to the phosphorylation of the GluR1 Ser831 site. In contrast, GluR2-lacking AMPARs participate in Ca^2+^-mediated excitotoxicity [106]. When TNFα is applied to hippocampal cell cultures, it can significantly increase the expression of GluR1 and the number of GluR2-lacking AMPARs (Figure 2); this effect can be reduced by sequestering TNFα [107]. TNFα can significantly increase presynaptic glutamate release in cultured neurons [105]. When applied to brain slices from neonatal mice, TNFα can cause dose-dependent neuronal excitotoxicity through increased calpains activity in the Purkinje neurons [108]. TNFα can also inhibit the activity of glutamate transporters and thereby increase neurotoxicity [109].

## 7. Anesthesia, Surgical Trauma, Glutamatergic Transmission, and Cognitive Dysfunction

Perioperative neurocognitive disorders are common in the elderly after surgery, especially in those with pre-existing diseases or frailty [110,111]. It can present as delirium with inattention and changes in the level of consciousness or a more delayed and subtle neurocognitive impairment, previously termed postoperative cognitive dysfunction (POCD) [58]. POCD can manifest as memory impairment, a decline in executive function, changes in mood and personality, or a combination of such [58]. The incidence of POCD in patients older than 60 years is 25.8% at 7 days postoperatively. Three months later, this value is still around 10% [112]. Although the pathogenesis of POCD is not yet clear, it may be related to the central cholinergic system, the excitatory amino acid system, and other neurotransmitters [113]. Animal and human studies suggest that neuroinflammation from surgery and anesthesia is important in its development [114,115].

Neuroinflammation, glutamatergic dysfunction, and cognitive dysfunction are intricately linked. Surgical trauma can incite a series of peripheral immune and inflammatory responses that result in profound peripheral inflammation [116,117], which can trigger neuroinflammation [118]. Among other factors, neuroinflammation contributes to the development of perioperative neurocognitive disorders, manifesting as acute delirium or more subtle delayed postoperative neurocognitive dysfunction. The latter has similar features to neurodegenerative diseases such as AD and may share similar pathophysiological mechanisms. Indeed, those with mild cognitive impairment or of advanced age are at particular risk of developing PNDs, and those who develop PNDs may later manifest more overt elements similar to AD.

Glutamate plays a vital role in long-term potentiation (LTP), a process that is considered to underpin learning and memory. Peripheral inflammation increases the expression of the NMDAR subunit 2B (NR2B) and NR2B receptor-mediated synaptic currents in the anterior cingulate cortex and contributes to pain sensitization [119]. Peripheral inflammation can affect the function of glutamate receptors and transporters and impair cognition. AMPARs are involved in excitotoxicity through the activation of NMDARs. The excessive stimulation of NMDARs or AMPARs can induce neuronal apoptosis [120,121].

In addition to the presence of peripheral and neuroinflammation, the perioperative picture is further complicated as most forms of surgery are usually performed under anesthesia. Volatile anesthetics have a significant impact on the levels of pro-inflammatory cytokines that, in turn, can affect glutamatergic transmission. Different anesthesia agents, anesthesia exposure times, ages, and surgical models have been evaluated and each combination can produce slightly different effects. The two most commonly clinically used volatile anesthetic agents, and hence evaluated in the most detail experimentally, are isoflurane and sevoflurane. The former has been in clinical use for a longer period but is increasingly being replaced by sevoflurane.

### 7.1. Isoflurane

Volatile agents have intrinsic effects on neurotransmission, especially on the glutamatergic system, either directly or via their effects on inflammatory cytokines. Brief exposure to isoflurane can significantly increase pro-inflammatory TNF-α, IL-6, and IL-1β levels [122]. Isoflurane can abate excitatory transmission by reducing the release and increasing the uptake of glutamate into the presynaptic terminal [123] and can reduce glutamate release in the hippocampus after ischemia [124].

Excitatory amino acid transporters have a significant role in glutamate reuptake at the synapses and, consequently, in cognition. Isoflurane can enhance the expression of EAAT3 on the plasma membrane via a protein kinase C (PKC)-dependent pathway [125,126] and imparts neuroprotective effects by preserving the function of EAAT3 for L-glutamate and L-cysteine uptake [127]. EAAT3 knockout mice have an obvious baseline cognitive impairment, and isoflurane anesthetic does not additionally affect the cognition of the mice [128]. Cao et al. also demonstrated that isoflurane inhibits context-related fear conditioning in EAAT3 knockout mice. In addition, increased GluR1 can be trafficked to the plasma membrane via EAAT3 [126]. The expression of EAAT1 can also be influenced by isoflurane; EAAT1 was shown to be increased in the neonatal rats after exposure to isoflurane anesthetic [129]. Isoflurane inhalation does not affect the activity of wild-type EAAT2 [130].

The effect of isoflurane on the brain is in part affected by the age at which the exposure takes place. During the development period of the brain, anesthetic exposure can interfere with the normal patterns of synaptogenesis and thus may impair the assembly of neural circuits, which in turn could affect cognition [131]. Calpain-2 is a neutral cysteine protease that is highly expressed in the CNS and can be activated by NMDARs [132]. In one study using neonatal mice, isoflurane exposure significantly increased the expression of the NR2B subunit compared with the NR2A subunit and the calpain-2 protease [133]. However, Wang et al. found that the expression of NR2A increased while NR2B decreased in the hippocampus of neonatal rats after the isoflurane exposure [129].

The effect of isoflurane exposure in adult rodents can also be quite variable. In 4 to 5-month-old mice, two hours of isoflurane inhalation can significantly improve cognitive performance and the expression of NR1 and NR2B subunits on the following day; however, this upregulation was only maintained for a relatively short period [134]. Lin et al. and Cao et al. both demonstrated that isoflurane could impair cognitive performance, as assessed by a Barnes maze and fear conditioning tests, in adult rodents after exposure [135,136]. The effect of isoflurane on cognition may be dependent on the dose and the duration of exposure. A shorter duration or lower concentrations of isoflurane induce some improvement in cognitive performance associated with increased NR2B expression. In contrast, a longer duration of exposure decreases NR2B expression and impaired cognition in adult mice [137]. In aged mice, isoflurane anesthesia can also impair cognitive function [135]. Isoflurane has also been shown to diminish learning and memory in older rats, accompanied by increased glutamate levels in the cerebrospinal fluid as well as GLAST expression in the hippocampus [138].

### 7.2. Sevoflurane

In a similar fashion to isoflurane, sevoflurane can also significantly increase the levels of cytokines that, in turn, can affect glutamatergic transmission. Sevoflurane can increase IL-6, IL-8, and TNFα by decreasing the activation of the PI3K/Akt/mTOR pathway in young rats [139]. Exposure to this drug can also directly reduce calcium-dependent glutamate release in the human brain [140]. In neonatal rats, sevoflurane exposure was shown to change LTP and long-term depression (LTD) by increasing the expression of GluR2-lacking AMPA receptors [141]. Exposure in utero increases the expression of NR2B, decreases the number of pyramidal neurons in the entorhinal cortex (ECT), and leads to abnormal development in the newborn [142]. Exposure during gestation also increases the expression of NR1 and NR2A in neonatal mice; however, in contrast, NR2B is decreased in the hippocampus [143]. Neonatal mice exposed to sevoflurane can have reduced activity of glutamatergic neurons in the amygdala. This leads to a learning deficit in fear conditioning after mature adulthood [144]. Sevoflurane can also induce glial dysfunction in neonatal rats and inhibit the glutamate-aspartate transporter (GLAST) through the Janus kinase/signal transducer and activator of transcription (JAK/STAT) pathway [145].

There appears to be age differences in the glutamatergic and cognitive responses to sevoflurane. A 2 h exposure to sevoflurane at a minimum alveolar concentration can improve cognitive performance and increase the expression of NMDA receptors 1 and 2B in 4–5-month-old mice [146]. On the other hand, two hours of sevoflurane exposure in 24-month-old rats was shown to result in impaired learning and memory [147]. Peng et al. also found that sevoflurane inhalation for 4 h caused cognitive impairment in 24-month-old but not in 3-month-old Sprague Dawley (SD) rats by inhibiting the PPAR-γ pathway. This impairment can be reversed by traditional Chinese medicine cistanches [147]. Sevoflurane exposure can also inhibit mGluRs and impair cognition. The mGluR-LTD in aged mice can increase the expression of small conductance calcium-activated potassium type 2 channels. The memory deficit can be reversed by a potassium channel blocker [148].

### 7.3. Surgical Trauma

The impact of general anesthesia on the immune system of healthy patients appears to be comparatively less substantial than that from surgical trauma, especially after major surgery [149,150]. Systemic inflammation, including that following trauma, is an evolutionary response to defend the body against infection or trauma [151], and the increased circulating cytokines have an effect on the glial cells and the glutamatergic system. The expression of TNF-α, IL-1β, IL-6, and IFN-γ fluctuates during the perioperative period [152]; for example, the expression of IL-6 is increased after surgery under isoflurane anesthesia [153].

The glutamatergic system also undergoes substantial changes in response to surgery. Both plasma and CSF glutamate levels are significantly increased in neurosurgical patients. Under physiological conditions, the free passage of plasma glutamate is inhibited by the intact blood–brain barrier; however, under pathological conditions, plasma glutamate levels have been shown to passively follow their gradient, traversing the damaged BBB to the cerebral extracellular space [154]. In 24 to 25-month-old rats, the increased expression of NR2 seen after abdominal surgery under isoflurane anesthesia correlated with cognitive impairment that could be attenuated by local or systemic analgesia. These results suggest that postoperative pain may have a role in cognitive impairment. However, this study did not distinguish the NR2 receptors’ subtype, which is a limitation [155].

As it appears that NR2 subunits participate in both pain perception and cognition, Zhang et al. showed that pain can significantly increase the levels of the pro-inflammatory cytokine TNF-α, increase the brain levels of cyclin-dependent kinase 5 (CDK5), and decrease the expression of NR2B in the medial prefrontal cortex without any changes in the hippocampus. In this study using a surgical incision on the paw of 9-month-old mice under isoflurane anesthesia, there was also hippocampus-independent learning impairment; however, all these changes in NR2B were attenuated by a CDK5 inhibitor or a local anesthetic agent. These results indicated that the surgical incision-induced nociception reduced the expression of NR2B by increasing the expression of CDK5 [156]. Riazi et al. used a model of colonic inflammation in SD rats—induced by the intracolonic injection of 2,4,6-trunitrobenzenesulfonic acid under isoflurane anesthesia—to show that the amplitudes of miniature excitatory postsynaptic currents increased without changing the frequencies and paired-pulse ratios, indicating changes in the postsynaptic effect. Furthermore, enhanced AMPA and NMDA mediated currents in the evoked excitatory postsynaptic currents (eEPSCs) in the Schaffer collateral pathway and the rectification index indicated an increased contribution from GluR2-lacking AMPA receptors [157].

Surgery can also alter the composition of AMPAR subunits, which is accompanied by impaired cognition. AMPAR GluR2 subunits were shown to be significantly decreased in the hippocampus after partial hepatic lobectomy under sevoflurane anesthesia in D-galactose-induced aging mice [158]. In addition, the internalization of GluR2 and cognitive impairment can be found after exposure to a high concentration of sevoflurane or propofol in a tibial fracture animal model; the mechanism underlying this effect may be related to a decrease in the activity of the PI3K-GluA2 pathway [159]. However, there exist different results obtained using various techniques. Electrophysiological recording in the CA1 hippocampus revealed a decrease in the AMPA/NMDA receptor ratio without a change in the rectification index in a tibial fracture model under sevoflurane anesthesia, indicating no internalization of the GluR2. This impairment could be reversed by the acetylcholinesterase inhibitor galantamine [160].

## 8. Potential Therapeutics

The perioperative period associated with major surgery can have profound impacts on the function of glutamate in the CNS. The relationship is particularly complex due to the multiple approaches by which surgery-induced inflammation can affect glutamatergic function, beginning with the assault on the blood–brain barrier. Once an array of peripheral cytokines and immune cells can gain entry to the CNS through the dysfunctional BBB, the ensuing neuroinflammatory response can affect glutamate release and reuptake, glutamate receptor phenotypes, and EAAT functions. Aberrant synaptic function and excitotoxicity in vulnerable regions contribute to cognitive dysfunction. The net effect may be further complicated by the intrinsic effects of volatile anesthetic agents on glutamatergic function that may be age- or duration-dependent.

It is perhaps this complicated interplay between such diverse factors peculiar to the perioperative period that has limited the translational potential of therapies targeting the glutamatergic system. However, there are a few promising experimental examples of agents improving cognitive function by suppressing inflammation that eventually alleviate the adverse changes related to glutamate handling and receptors. Dexmedetomidine is a sedative hypnotic drug that can decrease the level of pro-inflammatory cytokines and decrease the expression of NR2A and NR2B to protect against neuronal injury after exposure to sevoflurane [129,139]. Several herbal compounds have also shown experimental benefits. Senegenin is a component of the root *Polygala tenuifolia* that can improve cognition and the hippocampal expression of NR2B in sevoflurane-induced cognitive dysfunction [161]. Fingolimod (FTY720), a sphingosine-1-phosphate receptor modulator, can improve learning and memory and increase the expression of GluR2 after a partial hepatic lobectomy [158]. It can reduce T cell- and macrophage/microglia-mediated inflammation and attenuate astrocyte activation [162]. Vitexin is a bioactive compound extracted from hawthorn leaves that has a neuroprotective effect through the TRPV1 and NR2B pathways [163]. Interestingly, the β-lactam antibiotic ceftriaxone used to treat CNS infections can improve cognition by increasing GLT1 expression, thereby reducing neuroinflammation and apoptosis [164].

## 9. Conclusions

In summary, alterations to glutamate handling and glutamatergic transmission by neuroinflammation play a major role in the manifestation of perioperative neurocognitive disorders. We must build a more comprehensive picture of the differential effects of how each perioperative variable affects this system in order to develop more strategic therapeutic options.

## Figures and Tables

**Figure 1 biomolecules-12-00597-f001:**
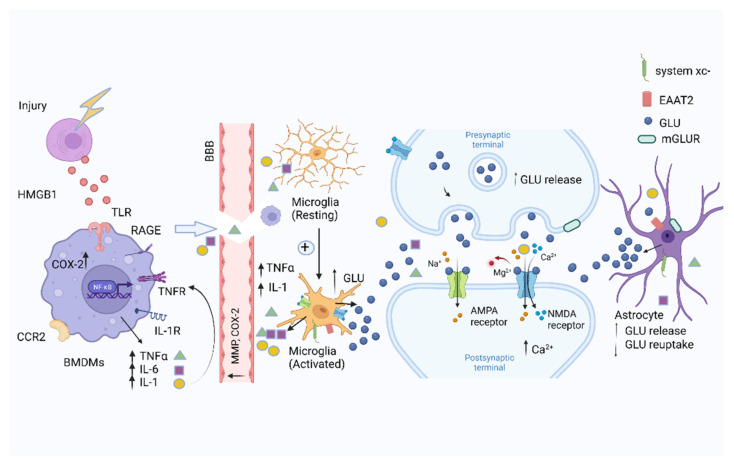
Systemic immune responses to trauma. Injured cells release DAMPs, including HMGB1, in response to surgical trauma after being combined with the TLR and RAGE, which can activate nuclear factor-kappa B (NF-κB) signaling pathways in BMDMs, promoting the release of pro-inflammatory cytokines, including IL-6 and TNFα, IL-1. The increased expression of COX-2 and MMPs disrupts the integrity of the blood–brain barrier. Pro-inflammatory cytokines activate microglia to further amplify the release of pro-inflammatory cytokines in the brain. Glia activated by the pro-inflammatory cytokines can further stimulate the release of glutamate. The postsynaptic intracellular Ca^2+^ concentration increases by the overactivation of NMDARs. The ability of astrocytes to clear glutamate is decreased. Figure created with Biorender.com (accessed on 2 March 2022).

**Figure 2 biomolecules-12-00597-f002:**
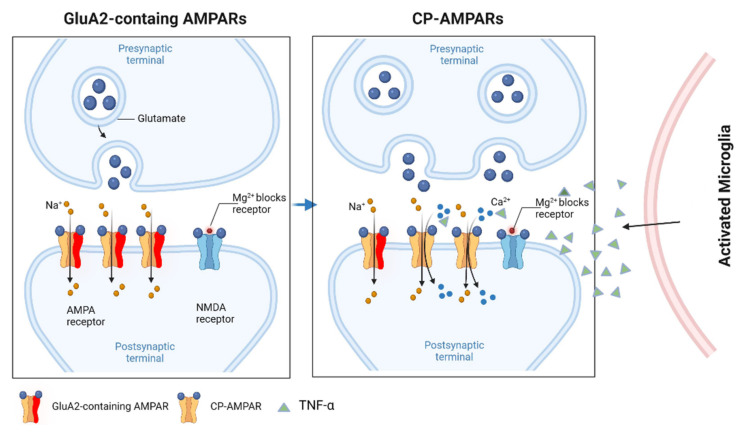
GluA2-lacking CP-AMPARs in neuroinflammation. GluA2-lacking receptors (CP-AMPARs) are relatively rare in most excitatory neurons in baseline conditions. However, GluA1/2 heteromers are replaced with GluA1 homomers after induction by pro-inflammatory cytokines, such as TNFα. The GluA2 heteromers are Ca^2+^ impermeable, whereas the GluA1 homomers (i.e., GluA2-lacking AMPARs) permit the passage of both Na^+^ and Ca^2+^. Figure created with Biorender.com (accessed on 2 March 2022).

## Data Availability

Not applicable.

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
