# Peer review of "Cerebral Glutamate Regulation and Receptor Changes in Perioperative Neuroinflammation and Cognitive Dysfunction"

_biomolecules, 2022, doi:10.3390/biom12040597_

Round 1
Reviewer 1 Report
The present review is an interesting study of the surgery effects on neuroinflammation and glutamatergic transmission, leading to cognitive impairments. The text it is well organized and reads well. However, there are several aspects, that may be improved:
- It is stated that the activity of glutamate it depends on NMDA, AMPA and glutamate transporters. Metabotropic glutamate transporters are missing here, and are important as it is stated after on the main text.
- Section 4.1. Within the second paragraph it is stated that glutamate activates microglia. However, it shouldn´t be cytokines the main activators of microglia after a surgical event, as is stated in figure 1 legend? Glutamate may activate microglia after glutamate transmission is affected, not as a first event after the surgery.
- Section 4. It is well described how microglia is activated after surgery, and the effects of this activation. However, astrocyte and oligodendrocyte response after injury it is not described. This decreases the reading flow of the section and reduces the importance of describing glutamate metabolism of these cells. Thus, this section may be improved
- Page 7 line 299. Impaired should be impair?
- Section 8. This is an interesting section describing potential neuroprotective agents for decreasing glutamate alteration after surgery. However, the title does not announce such interesting text. Thus, neuroprotective agents and main conclusions could be separated within different sections, to enhance the impact of describing potential therapeutic agents after surgery within the review?
Author Response
- It is stated that the activity of glutamate it depends on NMDA, AMPA and glutamate transporters. Metabotropic glutamate transporters are missing here, and are important as it is stated after on the main text.
Response: Thank you for this comment. Glutamate receptors can be divided into ionotropic and metabotropic glutamate receptors. Ionotropic glutamate receptors are primarily associated with fast synaptic transmission, while the metabotropic glutamate receptors are important for neuromodulatory effect. Glutamate is mostly uptaken by its transporter into the glial cells. Additional contents on the glutamate transporters are included in the revised manuscript.
- Section 4.1. Within the second paragraph it is stated that glutamate activates microglia. However, it shouldn´t be cytokines the main activators of microglia after a surgical event, as is stated in figure 1 legend? Glutamate may activate microglia after glutamate transmission is affected, not as a first event after the surgery.
Response:
Thank you for your comment and for pointing out a potential source of confusion. It is the activation of glutamate receptors on the microglia by the glutamate that is released by activated microglia and not activation of microglia by glutamate thus setting up a positive feedback loop. Glutamate does not directly activate microglia.
- Section 4. It is well described how microglia is activated after surgery, and the effects of this activation. However, astrocyte and oligodendrocyte response after injury it is not described. This decreases the reading flow of the section and reduces the importance of describing glutamate metabolism of these cells. Thus, this section may be improved
Response: Glial cells share some mechanisms of transcriptional regulation during neuroinflammation including NF-κB signaling and subsequent transduced signals. These pathways amplify the inflammatory activity of surrounding microglia and astrocytes. (Linnerbauer M et al, 2020, Neuron). Some information about astrocytes and oligodendrocytes activation are included in the revised manuscript.
- Page 7 line 299. Impaired should be impair?
Response: Thank you for your comment. It should be “impair” . We have modified it in the revised text.
5.Section 8. This is an interesting section describing potential neuroprotective agents for decreasing glutamate alteration after surgery. However, the title does not announce such interesting text. Thus, neuroprotective agents and main conclusions could be separated within different sections, to enhance the impact of describing potential therapeutic agents after surgery within the review?
Response: We thank the reviewer for this important comment. We have divided the therapeutics and the conclusions into different sections.
Reviewer 2 Report
This review by Zhang et al deals with an important topic: peri-/postoperative neuroinflammation as a possible explanation for postoperative cognitive dysfunction.
The review gives a short overview of the glutamatergic synapse, of the three classes of glial cells, of some leukocytes (macrophages and T-cells, but not neutrophiles and others), and some cytokines and their possible effects on glutamatergic neurotransmission.
Then they describe the effect of isoflurane and sevoflurane on glutamatergic neurotransmission before they describe possible effects of surgery (and postoperative pain) on cytokine levels and supposed glutamatergic parameters, quoting 7 studies altogether on this topic.
Evaluation:
Major point: For a review on postoperative cognitive dysfunction, the section on this topic is remarkably short, with very few studies referenced.
Minor points:
1. The Introduction contains many hypothetical statements presented as facts.
2. Introduction: Do the authors think that Alzheimer and Parkinson disease and MS are precipitated by a systemic immune response? This would have to be substantiated by more than a review article on neuroinflammation.
3. Glutamate is not “one of the most abundant excitatory neurotransmitters in the CNS”; it is the excitatory neurotransmitters in the CNS.
4. A lack of precision should be remedied, e.g.: “The calcium-permeable AMPARs (GluA2-lacking AMPAR) may emerge under some pathological conditions and neuroinflammation.” The authors refer to animal studies on status epilepticus – why not say so instead of using the blurry term “some pathological conditions and neuroinflammation”?
5. A lack of references underpinning their claims, e.g.: “Surgical trauma can incite a series of peripheral immune and inflammatory responses that result in profound peripheral inflammation [77] that in turn can trigger neuroinflammation.”
6. A speculative tendency: “The glutamatergic system is also undergoing substantial changes in response to surgery. Plasma glutamate level is significantly increased in neurosurgical patients [113].»
Plasma glutamate does not reflect glutamatergic neurotransmission in the CNS.
Author Response
Response to Review 2
This review by Zhang et al deals with an important topic: peri-/postoperative neuroinflammation as a possible explanation for postoperative cognitive dysfunction.
The review gives a short overview of the glutamatergic synapse, of the three classes of glial cells, of some leukocytes (macrophages and T-cells, but not neutrophiles and others), and some cytokines and their possible effects on glutamatergic neurotransmission.Then they describe the effect of isoflurane and sevoflurane on glutamatergic neurotransmission before they describe possible effects of surgery (and postoperative pain) on cytokine levels and supposed glutamatergic parameters, quoting 7 studies altogether on this topic.
Evaluation:
- Major point: For a review on postoperative cognitive dysfunction, the section on this topic is remarkably short, with very few studies referenced.
Response: Thank you for the suggestions. We have included some content introducing postoperative cognitive dysfunction in literature 101-107
Minor points:
- The Introduction contains many hypothetical statements presented as facts.
Response: Thank you for your comment. We have changed the language in this section and provide some additional references (please see our response to point 8 below)
- Introduction: Do the authors think that Alzheimer and Parkinson disease and MS are precipitated by a systemic immune response? This would have to be substantiated by more than a review article on neuroinflammation.
Response: Thank you for pointing out how our use of language could be misleading.
Systemic inflammation can result in neuroinflammation, mainly exhibited as microglial activation, production of inflammatory molecules, as well as recruitment of peripheral immune cells in the brain. Peripheral immune cells infiltrate into the brain through disrupted blood brain barrier and lead to neuroinflammation. Neuroinflammtion a risk factor Alzheimer's disease, Parkinson disease and MS. More references are included in the revised manuscript. We revised the text that inferred a definitive causal and temporal relationship between systemic immune response and the onset of these diseases.
- Glutamate is not “one of the most abundant excitatory neurotransmitters in the CNS”; it is the excitatory neurotransmitters in the CNS.
Response: Thank you for your comment. It is the major excitatory neurotransmitters in the mammalian CNS. (Y. Zhou, 2014, Journal of neural transmission ) and we have modified the text to reflect this
- A lack of precision should be remedied, e.g.: “The calcium-permeable AMPARs (GluA2-lacking AMPAR) may emerge under some pathological conditions and neuroinflammation.” The authors refer to animal studies on status epilepticus – why not say so instead of using the blurry term “some pathological conditions and neuroinflammation”?
Response: Thank you for your comment. GluA2-lacking AMPAR can be found in inflammation-related pathological conditions like epilepsy, glaucoma, drug addiction. We have added more details and some references in the revised manuscript
- A lack of references underpinning their claims, e.g.: “Surgical trauma can incite a series of peripheral immune and inflammatory responses that result in profound peripheral inflammation [77] that in turn can trigger neuroinflammation.”
Response: Thank you for your comment and we apologise for the oversight. References have been included in the revised maunscript.
- A speculative tendency: “The glutamatergic system is also undergoing substantial changes in response to surgery. Plasma glutamate level is significantly increased in neurosurgical patients [113].Plasma glutamate does not reflect glutamatergic neurotransmission in the CNS.
Response: Thank you for your comment. It should be both the plasma and CSF glutamate levels are significantly increased in the neurosurgical patients. Under physiological conditions, free passage of plasma glutamate is inhibited by the intact blood-brain barrier (BBB) as cerebral endothelial cells possess important anatomical features, saturable carrier systems and enzymes known to regulate the access of amino acids to the CNS. Under pathological conditions, however, plasma glutamate levels have been shown to passively follow their gradient, traversing the damaged BBB to the cerebral extracellular space. All these can reflect the glutamatergic system changes.
Round 2
Reviewer 2 Report
No further comments.
Author Response
I found the introductory section and the general part on glutamate somewhat inexact, for instance:
1.Lines 33-35: The wording indicates that astrocytes are the primary source of glutamate release, not neurons. It also appears that major alterations of ionotropic glutamate receptors alteration, and the consequences on glutamate transmission, are in astrocytes.
Response: Thank you for your comment. We modified the description that may have been misleading in the revised version.
2.Lines 37-39: glutamate clearance and the glutamine cycle mentioned here are confusing. These concepts are more precisely described below (lines 203-212).
Response: Thank you for your comment. We have deleted the confusing sentence.
3.Lines 75-77: mGluRs are mentioned here and again on lines 86-91 in paragraphs dealing with ionotropic receptors. Classification of mGluRs should be treated more in detail, and these receptors should be described in a dedicated paragraph at the end of this section.
Response: Thank you for your comment. We described mGluRs in more detail in another paragraph in the revised manuscript.
4.Lines 83-86: the authors claim that Ca2+-permeable AMPA receptors derive from GluA2-lacking receptors. One primary reason leading to Ca2+-permeable AMPA receptors is the altered editing of this subunit. MAPA receptor-containing the unedited GluA2(Q) subunit are Ca2+-permeable as well.
Response: In all AMPAR GluA1–4 subunit genes, there exists a conserved glutamine site at the second intramembrane domain that constitutes the inner face of the channel. At the mRNA level, this glutamine codon is edited to arginine, which confers channel resistance to calcium. mRNA editing selectively targets GluA2 subunits. (Heng-YeMan, 2011, Current Opinion in Neurobiology). Although CP-AMPARs can arise from either the lack of GluA2 or the presence of an unedited GluA2 in the receptor complex, the fact that less than 1% of all GluA2 RNA encodes unedited GluA2(Q) in the adult brain show that argues for the fact that most CP-AMPARs lack GluA2. (Yukio Kawahara, 2003, European Journal of neuroscience ), GluA2-lacking AMPARs are the composition of the majority of CP-AMPARs. We describe it more accurately in the revised manuscript.
5.Lines 117,118: the authors indicate that transporter reversal-mediated glutamate release occurs in endothelial cells, which release glutamate in the blood. There is no mention of the transporter reversal-mediated glutamate release from neuronal axon terminals and astrocyte perisynaptic moieties, which is of paramount pathological importance.
Response: Thank you for your comment. We have added some content about the glutamate released by the reverse transport in neurons and astrocyte in the revised manuscript.
6.Lines 196: Authors indicate that astrocytes regulate synaptic pruning. Synaptic pruning is a main activity of microglia rather than astrocytes, but this occurrence was not mentioned in the previous microglia-dedicated section.
Response: Thanks for your comment. We have added some content about the function of synaptic pruning by the microglia in the related paragraph.
